# The Impact of Metabolic Syndrome on Heart Failure in Young Korean Population: A Nationwide Study

**DOI:** 10.3390/metabo14090485

**Published:** 2024-09-04

**Authors:** Tae-Eun Kim, Do Young Kim, Hyeongsu Kim, Jidong Sung, Duk-Kyung Kim, Myoung-Soon Lee, Seong Woo Han, Hyun-Joong Kim, Hyun Kyun Ki, Sung Hea Kim, Kyu-Hyung Ryu

**Affiliations:** 1Department of Clinical Pharmacology, Konkuk University Medical Center, Seoul 05030, Republic of Korea; tekim@kuh.ac.kr; 2Division of Cardiology, Department of Internal Medicine, Ajou University Hospital and Ajou School of Medicine, Suwon 16499, Republic of Korea; sneeze@aumc.ac.kr; 3Department of Preventive Medicine, School of Medicine, Konkuk University, Seoul 05030, Republic of Korea; 4Division of Cardiology, Department of Medicine, Heart Vascular Stroke Institute, Samsung Medical Center, Sungkyunkwan University School of Medicine, Seoul 06351, Republic of Korea; jidong.sung@samsung.com (J.S.); dukkyung.kim@gmail.com (D.-K.K.); 5Department of Social and Preventive Medicine, Sungkyunkwan University School of Medicine, Suwon 16418, Republic of Korea; msnlee@skku.edu; 6Division of Cardiology, Dongtan Sacred Heart Hospital, Hallym University College of Medicine, Hwaseong 18450, Republic of Korea; hansw29@hanmail.net; 7Division of Cardiology, Department of Internal Medicine, Konkuk University Medical Center, Konkuk University School of Medicine, Seoul 05030, Republic of Korea; drkhj2000@kuh.ac.kr; 8Division of Infectious Diseases, Department of Internal Medicine, Konkuk University Medical Center, Konkuk University School of Medicine, Seoul 05030, Republic of Korea; kihkdr@kuh.ac.kr; 9Division of Cardiology, Hebron Medical Center, Phnom Penh 12406, Cambodia; khryumd@hanmail.net

**Keywords:** metabolic syndrome, heart failure, young age

## Abstract

Limited data are available regarding the effect of metabolic syndrome on heart failure (HF) development in young individuals. Utilizing data from the Korean National Health Insurance Service, we included a total of 1,958,284 subjects in their 40s who underwent health screening between January 2009 and December 2009 in Korea. Subjects were classified into three groups: normal, pre-metabolic syndrome (Pre-MetS), and metabolic syndrome (MetS). MetS was identified in 10.58% of males and 5.21% of females. The hazard ratio for HF in subjects with MetS was 1.968 (95% CI: 1.526–2.539) for males and 2.398 (95% CI: 1.466–3.923) for females. For those with Pre-MetS, the hazard ratio was 1.607 (95% CI: 1.293–1.997) in males and 1.893 (95% CI: 1.43–2.505) in females. Additionally, acute myocardial infarction and low hemoglobin levels were identified as significant risk factors for HF in both genders. MetS approximately doubled the risk of developing HF in individuals in their 40s. Pre-MetS was also a significant risk factor for HF in this population.

## 1. Introduction

Heart failure (HF) is a clinical condition that arises from structural or functional abnormalities of the heart [1]. HF affects over 26 million individuals globally, with population-based studies reporting that 1–2% of the worldwide population suffers from this condition [2,3,4,5,6]. Despite significant advances in therapies and prevention, HF remains associated with substantial healthcare expenditures and high rates of mortality and morbidity [7].

While the prevalence of HF is predominantly high in the elderly, recent studies indicate an increasing trend in the incidence of HF among younger populations [8,9]. According to a Danish report, the proportion of individuals under 50 years old experiencing incident HF increased from 3% in 1995 to 6% in 2012, even as the overall HF incidence rate had declined [9]. The reasons for this reverse trend in younger individuals remain unclear, but it is suspected to be related to the rising prevalence of metabolic disorders in the young population [10,11].

Although numerous studies have demonstrated the association between HF and metabolic syndrome (MetS) [12,13,14,15,16], most of the studies focused on the elderly or middle-aged population. The impact of MetS on HF in younger individuals has not been extensively studied. The different clinical phenotypes of HF in younger patients, as well as the limited evidence that suggests an association between HF and metabolic risk factors in these groups, highlight the need for a comprehensive study about the impact of MetS on HF development in young ages.

In this study, we investigated the influence of MetS on HF among a young population in their 40s by conducting a population-based study of 2 million individuals.

## 2. Materials and Methods

### 2.1. Database Source

This study utilized data from the National Health Insurance Service (NHIS) database, which is the universal health insurance system in South Korea. The NHIS covers more than 97% of the population and includes comprehensive information on patients’ demographics (age, sex, socio-economic variables, etc.), prescribed medications (generic drug names, prescription dates, duration, and routes of administration), and the utilization of medical care services (hospital admissions, outpatient visits, pharmaceutical visits, etc.). Additionally, the NHIS provides annual or biennial health screening examinations for individuals aged 20 years and older, which include lifestyle and behavior questionnaires, physical examinations, and blood tests. All diagnoses in the NHIS database are coded according to the International Classification of Diseases, 10th Revision (ICD-10). The data were de-identified and provided by the NHIS. This study was approved by the NHIS of Korea (No. NHIS-2020-1-538) and the Institutional Review Board of Konkuk University Medical Center (No. KUH 2020-07-096).

### 2.2. Study Population

Between January 2009 and December 2009, a total of 9,927,538 individuals in South Korea underwent health screening examinations. Of these, 7,969,254 participants were excluded based on the following criteria: (1) age not in their forties (younger than 40 years or older than or equal to 50 years), (2) a history of malignancy (ICD-10 codes C00.X-C99.X), or (3) a history of cardiovascular or cerebrovascular diseases, including atrial fibrillation (ICD-10 code I48), coronary artery disease (procedure codes M6561-4), myocardial infarction (ICD-10 code I21), heart failure (ICD-10 codes I42 or I50), cerebrovascular accidents (ICD-10 codes I60.X-I609.X), and peripheral arterial disease (ICD-10 codes I73 or I74) within five years prior to the screening. Ultimately, 1,958,284 individuals were included in the study. The screening period was from January 2004 to December 2008. During this time, the medical history of participants, including malignancy, cardiovascular disease, and cerebrovascular disease, was identified. The follow-up period started from the date of the national health check-up and continued until either the first occurrence of heart failure or 31 December 2016, whichever came first (Figure 1). These subjects were then categorized into three metabolic status groups based on the number of MetS components: normal group (0 components), pre-MetS group (1 or 2 components), and MetS group (3–5 components) (Figure 2).

### 2.3. Definitions of Variables

According to the modified criteria of the National Cholesterol Education Program (NCEP) Adult Treatment Panel III (ATP III), a diagnosis of metabolic syndrome (MetS) was made when at least three of the following five components were present: (1) abdominal obesity (waist circumference ≥ 90 cm for men, ≥85 cm for women); (2) elevated blood pressure (systolic BP ≥ 130 mmHg or diastolic BP ≥ 85 mmHg, or treatment for previously diagnosed hypertension); (3) elevated fasting glucose (≥100 mg/dL or treatment for previously diagnosed diabetes mellitus); (4) high triglycerides (≥150 mg/dL or drug treatment for high triglycerides); and (5) low high-density lipoprotein cholesterol (HDL-C) (<40 mg/dL for men, <50 mg/dL for women, or drug treatment for low HDL-C). Subjects with 3 to 5 MetS components were classified as having MetS, those with 1 or 2 MetS components were classified as pre-MetS, and those without any MetS components were classified as normal.

Body weight status was classified into five categories according to body mass index (BMI): underweight (BMI < 18.5 kg/m^2^), normal range (18.5 ≤ BMI < 23.0), upper normal (23.0 ≤ BMI < 25.0), overweight (25.0 ≤ BMI < 30.0), and obese (BMI ≥ 30.0). Smoking status was categorized as follows: (1) current smokers, defined as those who had smoked at least 100 cigarettes in their lifetime and continued smoking within one month prior to the 2009 national health check-up; (2) ex-smokers, defined as those who had smoked at least 100 cigarettes in their lifetime but had quit smoking at least one month before the 2009 health check-up; and (3) never-smokers, defined as those who had smoked fewer than 100 cigarettes in their lifetime. A family history of hypertension, diabetes mellitus, or stroke was determined based on questionnaire responses, identifying any immediate family member (parent or sibling) diagnosed with these conditions. Alcohol consumption frequency was classified as follows: (a) non-drinker, (b) 2–3 times per month, (c) 1–4 times per week, and (d) 5 or more times per week. Exercise frequency was categorized into three groups: (a) no exercise, (b) participation in vigorous (e.g., running, hiking, intense cycling) or moderate physical activities (e.g., brisk walking, tennis, moderate cycling) 1–4 times per week, and (c) 5 or more times per week. The biochemical and hematological results were categorized as follows: Total Cholesterol (mg/dL): total cholesterol levels were categorized into three groups: <200 mg/dL (considered within the normal range), 200–239 mg/dL (borderline high), and >239 mg/dL (high cholesterol). Alanine Aminotransferase (ALT) (IU/L): ALT levels were classified into three categories: <40 IU/L (indicative of normal liver function), 40–99 IU/L (mild to moderate elevation), and >100 IU/L (significantly elevated). Hemoglobin (g/dL): hemoglobin levels were categorized based on sex-specific reference ranges: <13.5 g/dL for males and <12 g/dL for females (defined as low hemoglobin, indicating anemia), 13.5–17.5 g/dL for males and 12–15.5 g/dL for females (considered within the normal range), and >17.5 g/dL for males and >15.5 g/dL for females (elevated levels, potentially indicating conditions such as polycythemia or dehydration). Creatinine (mg/dL): serum creatinine levels were categorized as ≤1.5 mg/dL and >1.5 mg/dL (indicative of impaired kidney function, suggestive of chronic kidney disease).

Quantitative variables (e.g., BMI, drinking status, biochemical results) were transformed into categorical variables to enhance the clinical interpretability of the results.

### 2.4. Primary Outcome and Follow-Up

The primary outcome of this analysis was the incidence of heart failure (HF) during the follow-up period. An HF event was defined by the occurrence of newly assigned ICD-10 codes for HF (I50) and any history of hospital admission. Follow-up began on the date of the health screening examination and ended at the incidence of HF, death, or 31 December 2016, whichever occurred first.

### 2.5. Statistical Analysis

The characteristics of the study subjects were presented using descriptive statistics. Chi-square tests were employed to compare baseline features across subjects with different MetS statuses. Cox proportional hazard models were utilized to estimate hazard ratios (HRs) and 95% confidence intervals (CIs) for the incidence of HF during the follow-up period. Before employing the Cox proportional hazard model, log–log survival curves were plotted to test the proportional hazard assumption. The initial models were unadjusted. Through stepwise addition of covariates, the final model was obtained by adjusting for demographic characteristics (age, smoking status, and exercise status), family histories of hypertension, stroke, and diabetes mellitus, body mass index (BMI), laboratory results, and the occurrence of acute myocardial infarction (AMI) during the follow-up period. In this model, age was included as a continuous variable, while smoking status, exercise status, and BMI were incorporated as categorical variables. When different categories of a categorical variable are included in the model, each category (except the reference category) is represented by a dummy variable. These dummy variables are entered into the model simultaneously to evaluate the effect of each category within the categorical variable.

All tests were two-sided, with a significance level of 0.05. All analyses were conducted using SAS version 9.4 (SAS Institute Inc., Cary, NC, USA). 

## 3. Results

### 3.1. Baseline Characteristics and Prevalence of Metablic Syndrome (MetS)

This study included a total of 1,363,999 males and 594,285 females. Among the males, 18.35% were classified as having MetS, 54.48% as Pre-MetS, and 27.17% as normal. For the females, 5.21% had MetS, 41.78% had Pre-MetS, and 53.01% were classified as normal. The prevalence of MetS varied significantly according to smoking status, alcohol use, and exercise frequency. Individuals with a family history of hypertension, diabetes, or stroke exhibited a higher incidence of MetS. Additionally, the prevalence of MetS was greater among individuals with elevated body mass index (BMI), total cholesterol, creatinine, and alanine aminotransferase (ALT) levels, as well as lower hemoglobin levels (Table 1).

### 3.2. Incidence Rates of Heart Failure by Metabolic Syndrome Status and Demographic Factors

MetS status was significantly associated with the incidence rate of HF. The incidence rates per 100,000 person-years in the MetS populations were 28.10 for males and 22.12 for females. In the pre-MetS populations, the rates were 14.24 for males and 13.47 for females. For the normal populations, the rates were 7.82 for males and 7.10 for females (Figure 3).

The baseline characteristics of individuals with HF compared to those without HF are presented in Table 2. Within the MetS group, HF was more frequently observed in both males and females than in the normal population. Although the proportion was smaller than that in the MetS population, the pre-MetS population also exhibited a higher incidence of HF compared to the normal population. In addition to MetS status, current smokers showed a higher proportion of HF compared to ex-smokers or non-smokers. BMI, ALT levels, hemoglobin levels, and a family history of DM were also related to the prevalence of HF. Alcohol consumption, total cholesterol levels, and family histories of hypertension or stroke were associated with HF prevalence only in males (Table 2).

### 3.3. Effect of Metabolic Syndrome and Pre-Metabolic Syndrome on Heart Failure

Unadjusted hazard ratios (HRs) of HF for MetS were 4.134 (95% CI: 3.322–5.144) in males and 3.421 (95% CI: 2.245–5.215) in females. For Pre-MetS, the HRs of HF were 2.057 (95% CI: 1.669–2.536) in males and 2.084 (95% CI: 1.591–2.73) in females. After adjusting for age, smoking status, alcohol consumption, exercise frequency, family histories of heart disease, HTN, DM, and stroke, as well as laboratory findings and the occurrence of acute myocardial infarction (AMI) during the follow-up period, both pre-MetS and MetS remained significant risk factors for HF. The adjusted HRs for HF associated with MetS were 4.134 (95% CI: 3.322–5.144) in males and 3.421 (95% CI: 2.245–5.215) in females. For Pre-MetS, the HRs of HF were 2.057 (95% CI: 1.669–2.536) in males and 2.084 (95% CI: 1.591–2.73) in females (Table 3). 

### 3.4. Additional Risk Factors for Heart Failure

The occurrence of AMI during the follow-up period significantly increased the risk of HF, with HRs of 238-fold in males and 214-fold in females. Low hemoglobin levels were also identified as risk factors for HF, with HRs of 1.695 (95% CI: 1.28–2.244) in males and 1.426 (95% CI: 1.072–1.898) in females. Other factors such as current smoking, ALT levels, total cholesterol levels, and family history of stroke showed sex-specific associations with HF risk. In males, a family history of stroke and elevated total cholesterol levels (>239 mg/dL) increased the risk of HF by 1.3-fold. In females, current smoking and slight elevation of ALT levels (40–99 IU/L) increased the risk of HF by 2.4-fold and 2.1-fold, respectively. BMI, creatinine levels, exercise frequency, alcohol consumption, increasing age within the forties, and family histories of heart disease, hypertension, or diabetes mellitus were not associated with the risk of HF (Table 3).

## 4. Discussion

In this study, we investigated the impact of metabolic syndrome (MetS) on heart failure (HF) in males and females in their 40s. The results indicated that MetS increased the risk of HF by 1.97-fold in males and by 2.40-fold in females. Pre-MetS increased the risk by 1.61-fold in males and by 1.89-fold in females. Additionally, acute myocardial infarction (AMI) and low hemoglobin levels emerged as significant risk factors for HF in this population.

Previous studies have shown that MetS is a significant risk factor for HF. For instance, a longitudinal study in the elderly reported that MetS increased the risk of HF by 1.58-fold [14]. Similarly, a study involving middle-aged men found an increased risk of HF by 1.8-fold [17]. Our prior research on middle-aged individuals demonstrated that MetS increased the risk of HF by 1.71-fold in males and by 2.14-fold in females [16]. In comparison, our current study focusing on individuals in their 40s showed even higher hazard ratios, suggesting that the impact of MetS on HF risk may be more prominent in younger populations compared to older individuals. Recently, Tromp et al. reported similar findings, indicating that younger people are more affected by metabolic problems such as obesity, hypertension, and diabetes, which contribute to HF [11]. The authors suggested that the increasing trend of HF among younger populations is associated with a rise in metabolic burden in this group. Contrary to the declining incidence of HF among the elderly, the prevalence of HF is increasing in younger populations, necessitating greater attention on this younger age population [9,10]. Therefore, early diagnosis and management of MetS are necessary to reduce the risk of HF in younger individuals.

Our study also indicated that the impact of MetS on HF is relatively higher in females than in males. Patients with heart failure with preserved ejection fraction (HFpEF) tend to be older and exhibit a twofold higher prevalence in females. Female hearts tend to be stiffer and exhibit a more pronounced tendency for increased left ventricular stiffness with aging compared to male hearts [15,18,19]. Obesity, a cornerstone of metabolic syndrome, appears to have stronger effects on diastolic dysfunction in women compared with men [20]. Endothelial dysfunction driven by obesity or metabolic syndrome is strongly associated with HFpEF syndrome through systemic inflammation. Unlike the inflammatory response in HFrEF, which results from cardiomyocyte damage, inflammation in HFpEF is triggered by a combination of extra-cardiac metabolic factors, including obesity, insulin resistance, and hypertension [21]. Given these facts, the authors speculated that the higher prevalence of HFpEF in females may contribute to the observed sex difference in the effect of MetS on HF.

Current smoking was a risk factor for HF in females. Given that smoking factors play an important role in the development of AMI, which increases the risk of HF, smoking cessation is essential for both sexes to improve the cardiovascular outcome [22].

In this study, BMI was not significantly associated with HF. While BMI is often used to determine obesity, it has limited value in evaluating actual body composition. BMI can misclassify individuals, particularly those experiencing a loss of muscle mass and an increase in fat deposition associated with aging, known as sarcopenic obesity [23]. Therefore, new markers of obesity, such as the visceral adiposity index, body composition analysis, and sarcopenic status assessment, may provide more accurate assessments of heart failure risk and its outcomes [24].

Several mechanisms may explain the connection between MetS and the onset of HF. The accumulation of visceral fat, a key factor in MetS, is associated with decreased levels of adiponectin and its receptors (types 1 and 2), leading to impaired insulin sensitivity and oxidative metabolism in individuals with MetS [25]. Insulin resistance further contributes to the functional and structural changes that result in myocardial injury and the development of HF [26]. In the context of insulin resistance, the myocardium shifts towards utilizing more free fatty acids instead of glucose, increasing its susceptibility to pressure overload and ischemic conditions [27].

Another mechanism involves the activation of the renin–angiotensin system. The renin–angiotensin system plays a central role in MetS, with enhanced activation of angiotensin II precursors, increased angiotensin II activity, and upregulated expression of angiotensin receptor 1 observed as a consequence of hyperglycemia and insulin resistance. At the myocardial level, increased angiotensin II activity contributes to oxidative stress, primarily through the activation of NADPH oxidase enzymes, leading to myocardial fibrosis, apoptosis, and ultimately, myocardial damage [28].

Recent studies have demonstrated that microRNAs play a significant role in the pathogenesis of heart failure. These studies have reported that microRNAs are involved in regulating signaling pathways, including MAPK, TGFβ, PI3K-Akt, PDGF, and IL-2, as well as pathways related to apoptosis, p53 activity, and angiogenesis. These pathways are closely linked to myocardial fibrosis, apoptosis, and ultimately contribute to myocardial injury [29]. Specific microRNAs, such as miR-222, have been shown to correlate with hyperglycemic parameters [30]. The miR-222/221 family has also been reported to be associated with the progression of heart failure [31,32]. Thus, microRNAs may serve as a potential missing link between metabolic syndrome and heart failure. However, further clinical studies are necessary to confirm these findings and to better understand their implications in clinical practice [31].

Our study has several limitations. Firstly, the retrospective design limits our ability to establish causality, and there may be bias introduced by unrecorded confounding variables. Secondly, the study population is restricted to individuals in their 40s from South Korea, which may limit the generalizability of the findings to other age groups or populations. Thirdly, lifestyle data were self-reported, which could introduce recall bias or inaccuracies. Fourthly, the number of new-onset heart failure was significantly lower in this given population, which may have limited statistical power.

## 5. Conclusions

This study investigated the impact of MetS on HF in individuals in their 40s using nationally collected real-world data. MetS increased the risk of developing HF by approximately twofold, with a greater effect observed in females than in males. Additionally, pre-MetS was identified as a risk factor for HF, although its impact was less pronounced than that of MetS.

## Figures and Tables

**Figure 1 metabolites-14-00485-f001:**
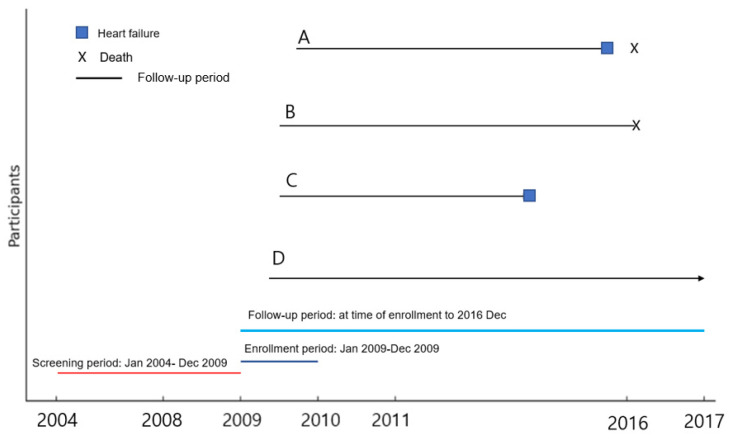
Timeline of screening, enrollment, and follow-up periods with participant outcomes. This figure illustrates the study’s screening period, enrollment period, and follow-up period. The follow-up period is explained through examples of hypothetical participants: Participant A enrolled in October 2009, was diagnosed with heart failure in September 2015, and subsequently died of heart failure in March 2016. The follow-up period for Participant A spans from October 2009 to September 2015. Participant B enrolled in June 2009 and died in a car accident in February 2016. The follow-up period for Participant B is from June 2009 to February 2016. Participant C enrolled in April 2009, was diagnosed with heart failure in 2014, and survived through December 2016. The follow-up period for Participant C is from April 2009 to 2014. Participant D enrolled in March 2009 and survived without any events through December 2016. The follow-up period for Participant D extends from March 2009 to December 2016.

**Figure 2 metabolites-14-00485-f002:**
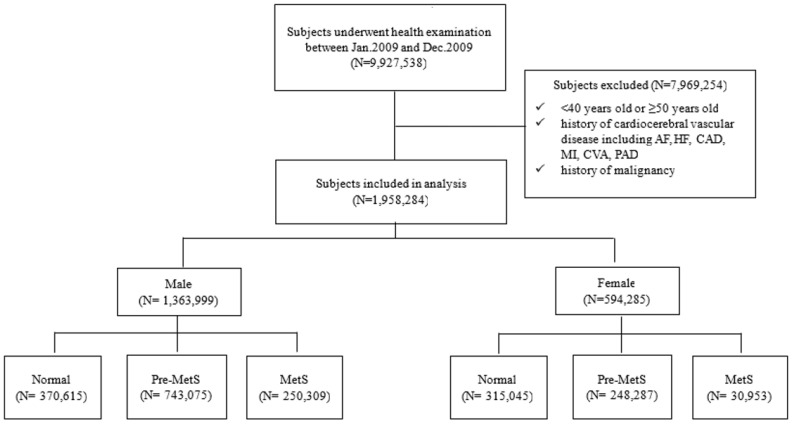
Flow diagram of the study. AF indicates atrial fibrillation; HF, heart failure; CAD, coronary artery disease; CVA, cerebellar vascular accident; PAD, peripheral artery disease; and MetS, metabolic syndrome.

**Figure 3 metabolites-14-00485-f003:**
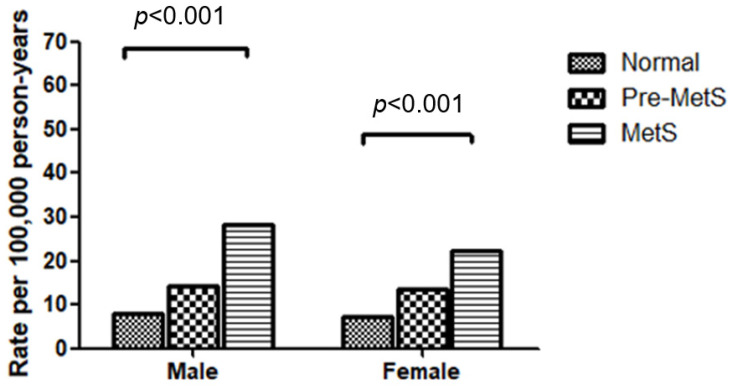
Incidence of heart failure according to metabolic syndrome status.

**Table 1 metabolites-14-00485-t001:** Baseline characteristics of study population according to metabolic syndrome status.

	Males				Females			
	Normal N = 370,615 (27.17)	Pre-MetS N = 743,075 (54.48)	MetS N = 250,309 (18.35)	*p*-Value	Normal N = 315,045 (53.01)	Pre-MetS N = 248,287 (41.78)	MetS N = 30,953 (5.21)	*p*-Value
**Smoking status**								
Non-smoker	108,206 (31.06)	186,630 (53.57)	53,559 (15.37)	<0.0001	292,085 (53.1)	229,944 (41.8)	28,067 (5.1)	<0.0001
Ex-smoker	72,032 (27.36)	144,365 (54.83)	46,911 (17.82)		8790 (54.2)	6502 (40.09)	926 (5.71)	
Current smoker	187,975 (25.25)	408,046 (54.8)	148,575 (19.95)		11,876 (49.43)	10,383 (43.21)	1769 (7.36)	
**Alcohol consumption**								
No drink	108,150 (30.88)	185,583 (52.99)	56,469 (16.12)	<0.0001	193,963 (52.16)	157,877 (42.45)	20,039 (5.39)	<0.0001
2–3 per month	208,989 (27.17)	419,430 (54.53)	140,730 (18.3)		103,714 (54.97)	75,983 (40.27)	8979 (4.76)	
1–4 per week	40,330 (21.22)	107,484 (56.54)	42,287 (22.24)		10,575 (50.06)	9292 (43.99)	1257 (5.95)	
≥5 per week	8098 (21.03)	22,170 (57.56)	8247 (21.41)		2046 (48.16)	1918 (45.15)	284 (6.69)	
**Exercise**								
No exercise	154,017 (26.61)	316,334 (54.66)	108,365 (18.73)	<0.0001	175,092 (52.96)	138,610 (41.93)	16,881 (5.11)	<0.0001
1–4 per week	82,087 (26.98)	165,111 (54.27)	57,034 (18.75)		62,212 (53.23)	48,634 (41.61)	6022 (5.15)	
≥5 per week	130,735 (27.92)	254,721 (54.41)	82,736 (17.67)		75,261 (52.8)	59,411 (41.68)	7868 (5.52)	
**Family history of hypertension**								
Yes	27,906 (21.25)	71,696 (54.6)	31,707 (24.15)	<0.0001	39,661 (48.91)	35,768 (44.11)	5662 (6.98)	<0.0001
No	237,651 (27.94)	462,524 (54.38)	150,405 (17.68)		173,966 (54.23)	131,137 (40.88)	15,667 (4.88)	
**Family history of diabetes mellitus**								
Yes	29,463 (22.45)	70,197 (53.49)	31,570 (24.06)	<0.0001	33,355 (48.17)	30,572 (44.15)	5317 (7.68)	<0.0001
No	236,013 (27.76)	463,762 (54.54)	150,480 (17.7)		180,136 (54.2)	136,232 (40.99)	16,003 (4.81)	
**Family history of stroke**								
Yes	14,615 (24.86)	32,194 (54.77)	11,975 (20.37)		13,702 (49.64)	12,125 (43.93)	1776 (6.43)	<0.0001
No	250,698 (27.19)	501,498 (54.39)	169,836 (18.42)		199,573 (53.42)	154,493 (41.35)	19,519 (5.22)	
**Total cholesterol (mg/dL)**								
<200	261,932 (33.5)	411,862 (52.68)	108,007 (13.82)	<0.0001	238,476 (55.15)	177,102 (40.96)	16,810 (3.89)	<0.0001
200–239	91,537 (20.99)	248,664 (57.03)	95,841 (21.98)		66,388 (49.98)	56,378 (42.44)	10,070 (7.58)	
>239	17,146 (11.73)	82,549 (56.48)	46,461 (31.79)		10,181 (35.03)	14,807 (50.95)	4073 (14.02)	
**ALT (IU/L)**								
<40	336,406 (32.08)	577,298 (55.05)	134,932 (12.87)	<0.0001	309,388 (53.86)	239,051 (41.61)	26,032 (4.53)	<0.0001
40–99	31,471 (11.08)	151,745 (53.44)	100,737 (35.48)		4992 (28.5)	8255 (47.13)	4267 (24.36)	
>100	2738 (8.72)	14,032 (44.67)	14,640 (46.61)		665 (28.91)	981 (42.65)	654 (28.43)	
**Hemoglobin (g/dL)**								
<13.5(Male), <12(Female)	23,528 (34.25)	36,674 (53.38)	8498 (12.37)	<0.0001	69,722 (52.48)	57,487 (43.27)	5639 (4.24)	<0.0001
13.5–17.5(Male), 12–15.5 (Female)	344,900 (27.02)	696,717 (54.58)	234,973 (18.41)		244,407 (53.23)	189,793 (41.34)	24,940 (5.43)	
>17.5(Male), >15.5(Female)	2187 (11.69)	9684 (51.76)	6838 (36.55)		888 (39.68)	980 (43.79)	370 (16.53)	
**Creatinine (mg/dL)**								
≤1.5	359,495 (27.29)	717,074 (54.44)	240,626 (18.27)	<0.0001	309,009 (53.02)	243,392 (41.76)	30,403 (5.22)	0.0387
>1.5	11,103 (23.75)	25,977 (55.56)	9671 (20.69)		6014 (52.54)	4884 (42.67)	549 (4.8)	
**Body Mass Index (** **kg/m^2^)**								
<18.5	15,245 (58.26)	10,561 (40.36)	363 (1.39)	<0.0001	33,371 (71.33)	13,244 (28.31)	170 (0.36)	<0.0001
18.5–22.9	195,445 (44.13)	228,356 (51.56)	19,056 (4.3)		215,994 (61.73)	129,301 (36.96)	4587 (1.31)	
23–24.9	100,361 (28.51)	213,364 (60.6)	38,353 (10.89)		44,400 (45.05)	49,312 (50.03)	4848 (4.92)	
25–29.9	59,083 (12.41)	269,099 (56.53)	147,815 (31.05)		20,766 (24.83)	48,734 (58.27)	14,129 (16.89)	
≥30	481 (0.72)	21,695 (32.43)	44,722 (66.85)		514 (3.33)	7696 (49.88)	7219 (46.79)	

Values are shown as number (percentage). The differences in the groups are presented as overall *p*-values. ALT indicates alanine transaminase.

**Table 2 metabolites-14-00485-t002:** Baseline characteristics of study population according to heart failure.

	Males			Females		
	Non-HF	HF	*p*-Value	Non-HF	**HF**	** *p* ** **-Value**
**Metabolic status**						
Normal	370,441 (99.95)	174 (0.05)	<0.0001	314,919 (99.96)	126 (0.04)	<0.0001
Pre-MetS	742,431 (99.91)	643 (0.09)		248,080 (99.92)	207 (0.08)	
MetS	249,865 (99.82)	444 (0.18)		30,907 (99.85)	46 (0.15)	
**Smoking status**						
Non-smoker	348,148 (99.93)	247 (0.07)	<0.0001	549,764 (99.94)	332 (0.06)	0.0001
Ex-smoker	263,133 (99.93)	175 (0.07)		16,205 (99.92)	13 (0.08)	
Current smoker	743,762 (99.89)	833 (0.11)		23,997 (99.87)	31 (0.13)	
**Alcohol consumption**						
No drink	349,854 (99.9)	348 (0.1)	<0.0001	371,642 (99.94)	237 (0.06)	0.9930
2–3 per month	768,511 (99.92)	637 (0.08)		188,558 (99.94)	118 (0.06)	
1–4 per week	189,897 (99.89)	204 (0.11)		21,110 (99.93)	14 (0.07)	
≥5 per week	38,457 (99.85)	58 (0.15)		4245 (99.93)	3 (0.07)	
**Exercise**						
No exercise	578,189 (99.91)	527 (0.09)	0.8200	330,371 (99.94)	212 (0.06)	0.7400
1–4 per week	303,949 (99.91)	282 (0.09)		116,799 (99.94)	69 (0.06)	
≥5 per week	467,748 (99.91)	444 (0.09)		142,445 (99.93)	95 (0.07)	
**Family history of hypertension**						
Yes	131,163 (99.89)	145 (0.11)	0.0124	81,037 (99.93)	54 (0.07)	0.8864
No	849,831 (99.91)	749 (0.09)		320,561 (99.93)	209 (0.07)	
**Family history of diabetes mellitus**						
Yes	131,085 (99.89)	145 (0.11)	0.0104	69,187 (99.92)	57 (0.08)	0.0492
No	849,509 (99.91)	745 (0.09)		332,167 (99.94)	204 (0.06)	
**Family history of stroke**						
Yes	58,703 (99.86)	81 (0.14)	<0.0001	27,578 (99.91)	25 (0.09)	0.0886
No	921,222 (99.91)	809 (0.09)		373,348 (99.94)	237 (0.06)	
**Total cholesterol (mg/dL)**						
<200	781,244 (99.93)	557 (0.07)	<0.0001	432,127 (99.94)	261 (0.06)	0.1453
200–239	435,622 (99.9)	419 (0.1)		132,743 (99.93)	93 (0.07)	
>239	145,871 (99.81)	285 (0.19)		29,036 (99.91)	25 (0.09)	
**ALT (IU/L)**						
<40	1047,786 (99.92)	850 (0.08)	<0.0001	574,118 (99.94)	353 (0.06)	0.0001
40–99	283,598 (99.88)	354 (0.12)		17,489 (99.86)	25 (0.14)	
>100	31,353 (99.82)	57 (0.18)		2299 (99.96)	1 (0.04)	
**Hemoglobin (g/dL)**						
<13.5(Male), <12(Female)	68,616 (99.88)	84 (0.12)	<0.0001	132,742 (99.92)	106 (0.08)	0.0119
13.5–17.5(Male), 12–15.5 (Female)	1,275,446 (99.91)	1143 (0.09)		458,870 (99.94)	270 (0.06)	
>17.5(Male), >15.5(Female)	18,675 (99.82)	34 (0.18)		2235 (99.87)	3 (0.13)	
**Creatinine (mg/dL)**						
≤1.5	1,315,989 (99.91)	1205 (0.09)	0.0673	582,435 (99.94)	369 (0.06)	0.3129
>1.5	46,696 (99.88)	55 (0.12)		11,437 (99.91)	10 (0.09)	
**Body Mass Index** (**kg/m^2^)**						
<18.5	2,6144 (99.9)	25 (0.1)	<0.0001	46,761 (99.95)	24 (0.05)	<0.0001
18.5–22.9	442,552 (99.93)	305 (0.07)		349,695 (99.95)	187 (0.05)	
23–24.9	35,1822 (99.93)	256 (0.07)		98,488 (99.93)	72 (0.07)	
25–29.9	475,453 (99.89)	543 (0.11)		83,554 (99.91)	75 (0.09)	
≥30	66,766 (99.8)	132 (0.2)		15,408 (99.86)	21 (0.14)	

Values are shown as number (percentage). The differences in the groups are presented as overall *p*-values.

**Table 3 metabolites-14-00485-t003:** Cox regression analysis performed to evaluate the association between clinical variables and heart failure.

	Males		Females	
Metabolic Status	Non-Adjusted HR (95% CI)	Adjusted HR (95% CI)	Non-Adjusted HR (95% CI)	Adjusted HR (95% CI)
Normal	1	1	1	1
Pre-metabolic synd.	2.057 (1.669–2.536)	1.607 (1.293–1.997)	2.084 (1.591–2.73)	1.893 (1.43–2.505)
Metabolic synd.	4.134 (3.322–5.144)	1.968 (1.526–2.539)	3.421 (2.245–5.215)	2.398 (1.466–3.923)
**Age, per year**	NA	1.067 (1.042–1.094)	NA	1.021 (0.98–1.064)
**Smoking status**				
Non-smoker	NA	1	NA	1
Ex-smoker	NA	0.924 (0.732–1.167)	NA	1.336 (0.683–2.611)
Current smoker	NA	1.139 (0.951–1.364)	NA	2.355 (1.503–3.69)
**Alcohol consumption**				
No drink	NA	1	NA	1
2–3 per month	NA	0.898 (0.764–1.055)	NA	0.995 (0.76–1.302)
1–4 per week	NA	1.134 (0.915–1.405)	NA	0.735 (0.368–1.47)
≥5 per week	NA	1.367 (0.967–1.932)	NA	0.283 (0.039–2.069)
Exercise				
No exercise	NA	1	NA	1
1–4 per week	NA	1.085 (0.913–1.29)	NA	0.850 (0.606–1.192)
≥5 per week	NA	1.059 (0.907–1.236)	NA	0.962 (0.715–1.295)
**Family history of heart disease**				
No	NA	1	NA	1
Yes	NA	0.925 (0.705–1.214)	NA	1.154 (0.709–1.877)
**Family history of hypertension**				
No	NA	1	NA	1
Yes	NA	0.975 (0.805–1.181)	NA	0.884 (0.644–1.212)
**Family history of diabetes mellitus**				
No	NA	1	NA	1
Yes	NA	1.100 (0.912–1.328)	NA	1.301 (0.958–1.768)
**Family history of stroke**				
No	NA	1	NA	1
Yes	NA	1.287 (1.018–1.628)	NA	1.431 (0.941–2.176)
**Body mass index**				
<18.5	NA	1.400 (0.814–2.409)	NA	1.278 (0.785–2.079)
18.5–22.9	NA	1	NA	1
23–24.9	NA	0.879 (0.718–1.077)	NA	1.177 (0.839–1.652)
25–29.9	NA	0.955 (0.789–1.157)	NA	1.136 (0.794–1.624)
≥30	NA	1.246 (0.943–1.647)	NA	1.308 (0.71–2.411)
**Hemoglobin (g/dL)**				
<13.5(Male), <12(Female)	NA	1.695 (1.28–2.244)	NA	1.426 (1.072–1.898)
13.5–17.5(Male), 12–15.5 (Female)	NA	1	NA	1
>17.5(Male), >15.5(Female)	NA	1.039 (0.694–1.556)	NA	1.401 (0.345–5.684)
**Creatinine (mg/dL)**				
≤1.5	NA	1	NA	1
>1.5	NA	1.235 (0.921–1.654)	NA	1.288 (0.635–2.611)
**Total cholesterol (mg/dL)**				
<200	NA	1	NA	1
200–239	NA	1.052 (0.898–1.231)	NA	0.980 (0.728–1.318)
>239	NA	1.281 (1.062–1.545)	NA	1.155 (0.702–1.9)
**ALT (IU/L)**				
<40	NA	1	NA	1
40–99	NA	0.964 (0.819–1.136)	NA	2.078 (1.287–3.357)
>100	NA	1.318 (0.94–1.849)	NA	0.804 (0.112–5.783)
**Acute myocardial infarction**				
No	NA	1	NA	1
Yes	NA	238.245 (203.671–278.688)	NA	214.404 (118.231–388.806)

## Data Availability

The data used to support the findings of this study can be made available by the corresponding author upon request.

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
