# Peer review of "The Impact of Metabolic Syndrome on Heart Failure in Young Korean Population: A Nationwide Study"

_metabolites, 2024, doi:10.3390/metabo14090485_

Round 1

Reviewer 1 Report

Comments and Suggestions for Authors

The authors presented the results on the the influence of metabolic syndrome on heart failure among young population in their 40s by conducting a population-based study of 2 million individuals. I have some comments:

1) In the Methods, list and describe in detail (with definitions, types) all indicators (demographic, social, clinical, etc.) that are included in the subsequent analysis to describe and compare the groups.

2) Describe in more detail the study design over time: dates of screening, variability of follow-up period, etc. Graphical representation on a timeline is welcome.

3) Discuss possible limitations of the study.

4) Explain in more detail the structure of regression models that include both binary, categorical and quantitative measures at the same time. And it is not clear why quantitative indicators (e.g. BMI) were transformed into categorical ones? And how were different categories of one categorical indicator simultaneously taken into account in the model?

5) I would like to see cumulative curves for the primary outcomes in groups.

Author Response

  1. Reviewer Comment: Provide a detailed explanation of all indicators included in the analysis to describe and compare the groups.

Response: Thanks for pointing this out. We also think that definitions of all indicators should be described in the method section. Therefore, we added paragraphs that describe definitions of demographic, social and clinical indicators in the method section

Page 3 line 109- 137 [Method]

Body weight status was classified into five categories according to body mass index (BMI): underweight (BMI < 18.5 kg/m²), normal range (18.5 ≤ BMI < 23.0), upper normal (23.0 ≤ BMI < 25.0), overweight (25.0 ≤ BMI < 30.0), and obese (BMI ≥ 30.0). [… ] Creatinine (mg/dL): Serum creatinine levels were categorized as ≤1.5 mg/dL and >1.5 mg/dL (indicative of impaired kidney function, suggestive of chronic kidney disease).

  1. Reviewer Comment: Provide more detailed information on the study design over time, including screening dates and variability in the follow-up period.

Response: We appreciate the reviewer's feedback and acknowledge the need to clarify the distinction between the screening period and the follow-up period. In our initial submission, we did not sufficiently differentiate these phases, which may have led to confusion. Therefore, we added following paragraph in the method section.

Page 3 line 88

The screening period was from January 2004 to December 2008. During this time, the medical history of participants, including malignancy, cardiovascular disease, and cere-brovascular disease, was identified. The follow-up period started from the date of the na-tional health check-up and continued until either the first

We also added following supplement figure to describe the timeline.

Supplement figure 1. Timeline of Screening, Enrollment, and Follow-Up Periods with Participant Outcomes. This figure illustrates the study's screening period, enrollment period, and follow-up period. The follow-up period is explained through examples of hypothetical participants:

Participant A enrolled in October 2009, was diagnosed with heart failure in September 2015, and subsequently died of heart failure in March 2016. The follow-up period for Participant A spans from October 2009 to September 2015. Participant B enrolled in June 2009 and died in a car accident in February 2016. The follow-up period for Participant B is from June 2009 to February 2016. Participant C enrolled in April 2009, was diagnosed with heart failure in 2014, and survived through December 2016. The follow-up period for Participant C is from April 2009 to 2014. Participant D enrolled in March 2009 and survived without any events through December 2016. The follow-up period for Participant D extends from March 2009 to December 2016.

  1. Reviewer Comment : Discuss possible limitations of the study.

Response: Thanks for your suggestion. We added possible limitations of the study in the discussion section

Page 15 line 306 4. Discussion

Our study has several limitations. First, the retrospective design limits our ability to es-tablish causality, and there may be bias introduced by unrecorded confounding variables. Second, the study population is restricted to individuals in their 40s from South Korea, which may limit the generalizability of the findings to other age groups or populations. Third, lifestyle data were self-reported, which could introduce recall bias or inaccuracies. Forth, the number of new-onset heart failure was significantly lower in this given popula-tion, which may have limited statistical power.

  1. Reviewer Comment

4-1. Explain in more detail the structure of regression models that include both binary, categorical and quantitative measures at the same time.

Response: Thank you for your insightful suggestion. We agree that the explanation of the structure of our regression models should be enhanced. As a result, we have added the following details to the Methods section to clarify our approach:

Page 4 line 156 2.5 Statistical analysis

In this model, age was included as a continuous variable, while smoking status, exercise status, and BMI were incorporated as categorical variables.

4-2. And it is not clear why quantitative indicators (e.g. BMI) were transformed into categorical ones?

Response: In this study, certain quantitative indicators, such as BMI, were transformed into categorical variables. This transformation allows the results to be more clinically meaningful. For example, discussing the relative risk of heart failure (HF) in "obese" individuals compared to those with "normal weight" provides clearer, more actionable information than interpreting risk per unit increase in BMI.

To clarify the rationale for this approach, we added the following sentence:

Page 4 line 139 2.3 Definitions of variablesof metabolic syndrome

Quantitative variables (e.g., BMI, drinking status, biochemical results) were transformed into categorical variables to enhance the clinical interpretability of the results.

4-3. And how were different categories of one categorical indicator simultaneously taken into account in the model?

Response : When different categories of a categorical variable are included in the model, each category (except the reference category) is represented by a dummy variable. These dummy variables are entered into the model simultaneously to evaluate the effect of each category within the categorical variable.

  1. Reviewer Comment: I would like to see cumulative curves for the primary outcomes in groups.

Response: We fully agree with your suggestion and recognize the value of providing cumulative curves for the primary outcomes in groups. However, accessing the data again would require re-approval from the Korean Health Insurance database, which involves a reauthorization process. Given the limited time available for preparing this revision, it has not been feasible to regenerate the data. We appreciate your understanding of this limitation.

Reviewer 2 Report

Comments and Suggestions for Authors

This study investigated the impact of MetS on HF in individuals in their 40s using real data collected at the national level. We also found that pre-metS is a risk factor for HF, but its effect is less pronounced than that of MetS. There are a few points that need to be mentioned:

1- The discussion section is relatively short and should be enriched with a comparison to the literature.

2- In the discussion section about the genetic predisposition of metabolic syndrome and heart failure, sharing information about the literature would contribute to the discussion section. For this purpose, "Is the microRNA-221/222 Cluster Ushering in a New Age of Cardiovascular Diseases. Cor et Vasa, 2023, 65.1: 65-67." Please use and refer to this study.

Author Response

Reviewer 2.

  1. Reviewer Comment: The discussion section is relatively short and should be enriched with a comparison to the literature.

Response: Thank you for the comment. We agree that the discussion is needed to be enhanced and we added current evidence regarding the relationship between metabolic syndrome and heart failure. To address this, we have added the following paragraph:

Page 14 line 280 4. Discussion

Several mechanisms may explain the connection between MetS and the onset of HF. The accumulation of visceral fat, a key factor in MetS, is associated with decreased levels of adiponectin and its receptors (types 1 and 2), leading to impaired insulin sensitivity and oxidative metabolism in individuals with MetS.[25] […]

At the myocardial level, increased angiotensin II activity contributes to oxidative stress, primarily through the activation of NADPH oxidase enzymes, leading to myocardial fibrosis, apoptosis, and ultimately, myocardial damage.[28]

  1. Reviewer Comment: In the discussion section about the genetic predisposition of metabolic syndrome and heart failure, sharing information about the literature would contribute to the discussion section. For this purpose, "Is the microRNA-221/222 Cluster Ushering in a New Age of Cardiovascular Diseases. Cor et Vasa, 2023, 65.1: 65-67." Please use and refer to this study.

Response: Thank you for the insightful suggestion. We agree that microRNAs play a significant role in both metabolic syndrome and heart failure, potentially serving as a crucial link between these conditions. In light of this, we have added the following paragraph to the discussion section and cited the recommended review article:

Page 15 line 295 4. Discussion

Recent studies have demonstrated that microRNAs play a significant role in the pathogenesis of heart failure. These studies have reported that microRNAs are involved in regulating signaling pathways, including MAPK, TGFβ, PI3K-Akt, PDGF, and IL-2, as well as pathways related to apoptosis, p53 activity, and angiogenesis. These pathways are closely linked to myocardial fibrosis, apoptosis, and ultimately contribute to myocardial injury.[29] Specific microRNAs, such as miR-222, have been shown to correlate with hyper-glycemic parameters.[30] The miR-222/221 family has also been reported to be associated with the progression of heart failure.[31,32] Thus, microRNAs may serve as a potential missing link between metabolic syndrome and heart failure. However, further clinical studies are necessary to confirm these findings and to better understand their implications in clinical practice. [31]

Reference

  1. Askin L, Tanriverdi O. Is the microRNA-221/222 Cluster Ushering in a New Age of Cardiovascular Diseases? Cor Vasa. 2023;65(1):65-67. doi: 10.33678/cor.2022.050.

Round 2

Reviewer 1 Report

Comments and Suggestions for Authors

The article has been much improved, but I still have comments:

1) I strongly recommend moving the figure from the Supplement to the main text of the Methods, as I believe it is an important element of the paper.

2) Sentences from Response 4-3 (“When different categories of a categorical variable are included…”) must be added to Methods (possibly with modification).

Author Response

1) I strongly recommend moving the figure from the Supplement to the main text of the Methods, as I believe it is an important element of the paper.

Response : Thank you for your valuable feedback. As per your suggestion, We have incorporated the following sentences into the method section section (page 3 line 97)

2) Sentences from Response 4-3 (“When different categories of a categorical variable are included…”) must be added to Methods (possibly with modification).

Response : I agree with your suggestion, and I believe it enhances the readability of the manuscript for the readers. The previously mentioned sentences have been added to the Methods section accordingly.